# Exploring Support Provided by Community Managed Organisations to Address Health Risk Behaviours Associated with Chronic Disease among People with Mental Health Conditions: A Qualitative Study with Organisational Leaders

**DOI:** 10.3390/ijerph19095533

**Published:** 2022-05-02

**Authors:** Julia Dray, Lauren Gibson, Tara Clinton-McHarg, Emma Byrnes, Olivia Wynne, Kate Bartlem, Magdalena Wilczynska, Joanna Latter, Caitlin Fehily, Luke Wolfenden, Jenny Bowman

**Affiliations:** 1School of Psychological Sciences, College of Engineering, Science and Environment, University of Newcastle, Callaghan, NSW 2308, Australia; julia.dray@uon.edu.au (J.D.); olivia.wynne@newcastle.edu.au (O.W.); kate.bartlem@uon.edu.au (K.B.); joanna.latter@newcastle.edu.au (J.L.); caitlin.fehily@newcastle.edu.au (C.F.); jenny.bowman@newcastle.edu.au (J.B.); 2Priority Research Centre for Health Behaviour, University of Newcastle, Callaghan, NSW 2308, Australia; tara.clinton-mcharg@newcastle.edu.au (T.C.-M.); luke.wolfenden@newcastle.edu.au (L.W.); 3Hunter Medical Research Institute, New Lambton Heights, NSW 2305, Australia; emma.byrnes@newcastle.edu.au; 4The Australian Prevention Partnership Centre, Sax Institute, Ultimo, NSW 2037, Australia; wilczynska.magdalena1@gmail.com; 5School of Medicine and Public Health, College of Health, Medicine and Wellbeing, University of Newcastle, Callaghan, NSW 2308, Australia; 6Hunter New England Population Health, Hunter New England Local Health District, Wallsend, NSW 2305, Australia

**Keywords:** community managed organisations, health risk behaviours, mental health conditions, barriers and facilitators, care provision

## Abstract

People living with mental health conditions experience a significantly reduced life expectancy compared to people without, largely linked to health risk behaviours and associated chronic disease. Community managed organisations (CMOs) represent an important setting in which to address health risk behaviours among people with mental health conditions. However, little is known about how these behaviours (smoking, poor nutrition, alcohol consumption, inadequate physical activity, poor sleep: SNAPS) are being addressed in this setting. One-on-one, semi-structured telephone interviews were conducted with a sample of 12 senior staff, representing 12 CMOs in New South Wales, Australia to: (1) explore types of support provided by CMOs to address the SNAPS behaviours of consumers living with a mental health condition; and (2) assess perceived organisational and staff level barriers and facilitators to providing such support. Transcribed interviews were analysed using inductive thematic analysis. This study found there was a range of supports offered by CMOs, and these differed by health risk behaviour. Findings suggest CMOs are well-placed to embed SNAPS supports as a part of their service provision; however, available funding, consistency of supports, workplace policies and culture, collaboration with other available supports, staff training and education, all impacted capacity.

## 1. Introduction

Globally, people living with mental health conditions have a significantly reduced life expectancy compared to people without mental health conditions (10–30 years worldwide) [1,2]. This gap in life expectancy is largely attributable to a higher likelihood of engagement in health risk behaviours (e.g., smoking, poor nutrition, alcohol consumption, inadequate physical activity, and poor sleep—SNAPS) and associated higher rates of chronic disease (such as cancer, cardiovascular disease, and respiratory diseases) [2]. To reduce this chronic disease burden, targeting health risk behaviours among this population is recommended across mental health services [2,3,4,5].

There has been a growing recognition of the importance of community managed organisations (CMOs) for delivering support to people with mental health conditions [6,7,8,9]. Internationally, CMOs are sometimes referred to as third sector, non-government, or not-for-profit organisations [6,10]. CMOs generally utilise a trauma-informed, recovery-oriented practice approach which is a consumer-led model of care. They provide a diversity of supports including those that address: psychosocial disability, rehabilitation and clinical care, accommodation support and outreach, employment and education, self-help and peer-support, promotion and advocacy, information provision, helpline and counselling services, and leisure and recreation [11]. CMOs have also been identified as a potentially important setting to address the health risk behaviours that impact physical health, as well as the mental wellbeing, of people living with mental health conditions [5,7,8].

Research exploring care provision addressing health risk behaviours within CMOs for consumers with mental health conditions has been limited by: (1) focusing on physical health broadly (i.e., encompassing all health from minor ailments through to chronic diseases such as cancer, with no specific focus on health risk behaviours) ([12,13,14,15]); (2) focusing on one health risk behaviour only [16,17]; (3) combining results for CMO-delivered and public mental health services [18,19,20,21]; and (4) combining results for populations of individuals with and without mental health conditions [22,23]. Outside of Australia, no research to the authors knowledge has explored the care being provided by CMOs to address multiple health risk behaviours among consumers with mental health conditions. One study in the United States evaluated a program aimed at better addressing the physical health of people with mental health conditions (through screening, monitoring, and coordinating services) where health risk behaviours were considered in the program evaluation [24]. This study found that just over half of sites implementing the program (56%) reported having an electronic registry with information about consumers’ health risk behaviour needs and services used to address these behaviours [24], though did not identify which health risk behaviours were considered or whether these were addressed equally.

Within Australia, a study conducted by Gibson et al. [25] explored CMO staff provision of care to address multiple health risk behaviours for consumers with mental health conditions in a CMO setting. The study identified variable care provision depending on the risk behaviour (e.g., alcohol consumption) and type of care provided (i.e., ask, assist, connect) [25]. In another study by Gibson et al. [7] where surveys were undertaken with CMO leaders, it was found that care provision for physical activity and nutrition was more commonly provided compared to care provision for alcohol consumption, tobacco smoking, and sleep. Further, the presence of organisational level guidelines for preventive care provision and staff training in preventive care delivery was associated with health risk behaviour screening and providing support to consumers for all five health risk behaviours [7].

This previous quantitative research was able to provide insight into the level of care provision for health risk behaviours for consumers with mental health conditions. However, there remains an important gap in the literature regarding how CMOs may be specifically addressing each of the five health risk behaviours and the factors that enable or impede the provision of preventive care from an organisational perspective. Such understanding is important to inform future strategies to increase the capacity of CMOs to provide preventive care, and more detailed accounts may be gained through qualitative research.

Therefore, the aim of this study was to conduct qualitative interviews with leaders from a range of CMOs in NSW to comprehensively explore:(1)The type of support provided to address five key health risk behaviours (smoking, poor nutrition, alcohol consumption, inadequate physical activity, and poor sleep) among CMO consumers with a mental health condition, and(2)The perceived organisational and staff level barriers and facilitators to providing support to address health risk behaviours.

## 2. Materials and Methods

### 2.1. Design and Setting

One-on-one semi-structured telephone interviews [26] with standardised open-ended questions [27] were conducted with a convenience sample of senior management staff, who had been invited to take part in a larger quantitative survey of CMOs (*n* = 85) in NSW, Australia. The study was approved by the University of Newcastle Human Research Ethics Committee (Approval number: H2018/0354).

#### Setting Context

The NSW CMO sector consists of a range of organisations with diverse characteristics in terms of organisational scale including number of locations, staff, and consumers, and the type of support provided [7]. CMOs also vary in how specialised they are for different populations of consumers with mental health conditions (e.g., by age, mental health condition, etc.) [5].

Organisational funding for CMOs is available through a diverse range of sources including Commonwealth, State, and Local Governments, as well as sponsorship, donations, and philanthropic grants [5]. NSW programs such as the Housing and Accommodation Support Initiative (HASI), and Community Living Supports (CLS) are currently provided by 8 and 7 CMOs in NSW, respectively [28,29]. These programs provide consumers with daily living support and capacity building skills (shopping, managing finances, accessing transport), attending mental and physical health appointments, gaining education and employment opportunities, social inclusion, and accessing other supports, for example, through The National Disability Insurance Scheme (NDIS) [28,29]. The NDIS provides individual funding to consumers with a psychosocial disability to access CMO services [30] for supports tailored to individual need, such as daily personal activities, transport to enable community and social activities, assistance with household tasks and home modifications [30]. Additional programs for people with mental health conditions are managed by Primary Health Networks (PHNs; independent primary health care organisations [31]) and consist of the National Psychosocial Support Measure and Continuity of Support funding [32].

### 2.2. Participants and Recruitment

#### 2.2.1. Community Managed Organisations

Organisations invited to participate in the current study were identified from among CMOs (*n* = 85) who were invited to take part in an online survey conducted by the research team [7]. Organisations had been considered eligible for participation in the online survey if they identified as a CMO that provided care to adults (18 years of age or older) with mental health conditions (or their families or carers) in NSW. The procedure for identifying potentially eligible organisations for participation in the original online survey included use of an online directory of mental health services in NSW [33], with subsequent checking as necessary via web searches, phone calls, and the member list of the NSW peak body for CMOs to clarify eligibility. Details of how the sample was identified have been reported elsewhere [7].

#### 2.2.2. Organisational Leaders

Organisational leaders of eligible CMOs were able to participate if they: were aged 18 years or over, employed in a senior management role (e.g., CEO or Area Manager), or staff who had been nominated by senior management as an appropriate proxy. Organisational leaders were chosen as appropriate key informants for the study due to their broad understanding of how their CMO operates across all service locations, rather than just within one particular service site. Further, members of senior management are often responsible for all aspects of a CMO’s functioning, and are important stakeholders for promoting, adopting, and overseeing the governance and implementation of organisational initiatives.

CMO leaders who represented the 85 organisations invited to take part in the previous study were contacted by the research team via several methods, sequentially. First an email invitation, next a postal mail invitation, and lastly up to three telephone attempts were made to invite participation in the study. CMO leaders indicating interest in participating were provided with a participant information statement via email and an agreed interview time was scheduled. Informed verbal consent was obtained prior to commencement of each interview.

### 2.3. Measures

An interview guide for the semi-structured interviews was developed to identify how organisations may be addressing five key health risk behaviours (tobacco smoking, nutrition, alcohol consumption, physical activity, and poor sleep—SNAPS) among CMO consumers with mental health conditions and to explore the potential barriers and facilitators to delivering this care (see Appendix A). The interview guide was pilot tested with a senior CMO staff member who did not take part in the study.

For each health risk behaviour in turn, the interview questions explored the support provided by staff to address that behaviour, and factors (at an organisational and staff level) that hindered or facilitated the provision of this support (see Appendix A for full questions). Organisational level factors were defined as those operating at a senior management level that subsequently impacted the frontline care provision (e.g., developing action plans and goals for the organisation regarding health risk behaviour support, and planning and conducting evaluations of the support provided). Staff level factors were defined as those operating on a day-to-day level (e.g., staff having access to information and resources regarding health risk behaviour support). For the purpose of the interview, participants were asked to only consider the health risk behaviour support provided to adults (18 years of age or older) with a mental health condition.

### 2.4. Data Collection

All interviews were conducted via telephone (and audio recorded) by one member of the research team (LG) who had previous experience conducting qualitative research interviews. Before each interview, the interviewer confirmed all participants had received, read, and understood the information statement, and that any questions had been answered. Recruitment continued until meaning saturation (i.e., rich ideas and issues arising from the interviews) was reached and no additional information was being gained from the interviews [34]. Audio recordings were transcribed verbatim by a professional transcription service external to the research team bound by confidentiality agreement.

### 2.5. Data Analysis

Inductive thematic analysis [35] was used to guide theme generation. Each transcribed interview was coded by two of three independent coders (EB, MW, and MR), using NVivo version 12 software. Once coding was complete, coders met and had in-depth discussions to generate the patterns of shared meaning to develop themes.

## 3. Results

From the 85 invited eligible CMOs, a total of 12 participants from 12 CMOs agreed to participate and completed interviews (14%) between March and July 2020. The interviews ranged in time from 40 min to 2 hours (average = 67.8 min). Two participants were employed as Chief Executive Officers and the remaining 10 participants were employed in other senior management roles (e.g., ‘NSW State Manager’ and ‘Executive Manager, Operations’).

Qualitative analysis generated two major themes (see Figure 1). The first theme related to types of support that CMOs provide consumers to address SNAPS, with three sub-themes identified: (1) practical education and life skills, (2) referral pathways and local connections, and (3) staff encouragement and peer support. The second theme represented what allowed CMOs to address consumer SNAPS, with 5 sub-themes identified; two predominately spoken about as barriers (lack of funding, and perceived limited ability to influence behaviours) and three as facilitators: (workplace policies and culture, collaboration with available supports, and staff education and training).

### 3.1. Types of Support Provided by CMOs to Address SNAPS Behaviours for Consumers with a Mental Health Condition

The analysis generated three main types of SNAPS supports to consumers including: practical education and life skills, referral pathways and local connections, and staff encouragement and peer support (see Figure 1).

#### 3.1.1. Practical Education and Life Skills

CMO leaders discussed the importance of providing education and supports that were practical and promoted ‘take away’ life skills for each SNAPS behaviour among consumers. Many CMOs spoke of the availability of educational resources (e.g., websites, pamphlets), practical supports (e.g., workshops on the financial impacts on smoking smoking) and Nicotine Replacement Therapy (NRT) in helping consumers to quit or reduce smoking.


*We’ve got our website, digital health website and it’ll have a section dedicated to smokers. It just asks whereby these people to just reflect on their smoking, whether they smoke or not, and whether they’d like information or support to quit.*
CMO leader 2


*We support them to get things like nicotine patches and [gum].*
CMO leader 5

Similarly, practical educational nutrition lessons were a popular support discussed by CMOs. For example, cooking lessons, food shopping, meal planning, and ingredient label reading were some of the supports offered to consumers that provided hands on nutrition skills.


*We have services develop resources, like cookbooks. Cooking for one, cookbooks, so, how to make the most of a supermarket shop and budget and make a large number of meals out of one shop, with pictures. All of the steps. All the ingredients.*
CMO leader 2


*There is also a program called [name of nutrition program]...that takes people through the cycle of growing or understanding about the nutritional balance and priorities and then looking at the cycle of that from growing things to preparing and eating things together as a group.*
CMO leader 3

Some CMO leaders discussed in-house practical education and informative sessions in supporting consumers to reduce their alcohol consumption.


*The [physical health checklist activity] the [form name] and the website all [explore] alcohol, and we do ask the question directly…if people would like to reduce it. If they’ve got concerns about the amount of alcohol…The website links people to information, being able to reduce.*
CMO leader 2


*We have an hour and a half session on lifestyle practices which is around reducing alcohol consumption, understanding even what healthy alcohol consumption is. Yes, we educate people because it comes as a huge shock to most of them to recognise what two standard drinks are.*
CMO leader 7

CMO leaders expressed the importance of supporting improved levels of physical activity among consumers. CMO leaders referred to both in-house and external physical activity programs and education sessions to improve this health behaviour among consumers including: structured exercise programs, group fitness activities, informal walking groups, and educational resources.


*It’s a 12-week intensive program for people with mental illness to—it’s focused on the cardio-metabolic syndrome, trying to get people fitter, lose weight…We provide gym memberships to people that join that.*
CMO leader 1


*We do have a strong focus on [physical activity] when we hold events as an organisation, like we have regular picnic days, we have—and have had an annual volleyball.*
CMO leader 2

Finally, some CMO leaders discussed lifestyle education sessions involving sleep hygiene and sleep routines as additional supports offered to consumers to improve their sleep behaviours.


*One of the quarterly health promotion topics. People talk about things like having an evening routine, for example, and ways to help you drift off to sleep. Maybe it might be mindfulness or meditation or different apps. It’s strategies to overcome those.*
CMO leader 2

#### 3.1.2. Referral Pathways and Local Connections

CMO leaders discussed the importance of supporting consumers where possible by incorporating referral to in-house supports, or through referring on to existing local professionals and services outside their organisation; again, with variation by health behaviour. For example, where there was no capacity within the organisation to support smoking cessation, CMO leaders discussed outsourcing to pre-existing services including the NSW Quitline and Cancer Council initiatives, as well as referrals to GPs, counsellors, psychologists, and Local Health District (LHD) services.


*We would probably issue them with the 13 QUIT Line number and refer them to their GP for appropriate treatment.*
CMO leader 11

For nutrition, some CMOs offered in-house programs that involved support from dieticians and nutritionists, while other CMOs spoke of referrals to health professionals outside their organisation such as nutritionists, dieticians, and speech pathologists.


*[Name of nutrition program] that’s running at the moment, [addresses] nutrition and there’s a dietician involved there. A nutritionist and a dietician, which help and support the consumers.*
CMO leader 1


*If it leaned to eating disorder territory, we would look at…connecting with a specialist.*
CMO leader 6

One CMO leader discussed connection to food providers to support consumers’ nutritional needs as an additional support.


*If we notice someone’s not cooking or not eating as well as they could, we will look at things like [name of food delivery service] as an option.*
CMO leader 12

All 12 CMO leaders spoke of the referral pathways in place in relation to supporting consumers to reduce their alcohol consumption. Many CMO leaders felt that practitioners including alcohol and other drugs services, GPs, and counsellors were best placed to offer support for behaviour change related to alcohol consumption.


*If we thought somebody had a serious alcohol problem, we would most likely, with their consent, refer them to an alcohol and drug counsellor or an alcohol and drug specific service.*
CMO leader 1


*We would have to refer them on to another organisation to get support with that. It would just be the case manager finding out who’s local and who can support and linking them up.*
CMO leader 9

In relation to improving consumers’ physical activity levels, some CMO leaders discussed the importance of working with exercise physiologists and personal trainers.


*The staff at all the different [name of group exercise centres] have been great. So that’s one we’ve really tapped into. Then where there’s not a [name of group exercise centre], we might go to a [name of gym] or whatever. They’ve been really open and really helpful. A lot of the times, they’ll give us a free personal trainer to work with the clients and staff. They’ve really—I must say, really embraced it.*
CMO leader 4

Finally, not all CMO leaders discussed that their service provided sleep supports. CMOs that did provide such supports often made connections to pre-existing medical services, particularly when medication was involved.


*It could be related to tiredness, which could be a medication—so it might be saying, maybe talk to your doctor about the timing of medications.*
CMO leader 2

#### 3.1.3. Staff Encouragement and Peer Support

Staff encouragement was discussed in relation to all SNAPS behaviours; however, it was most frequently spoken about in relation to physical activity. Motivating consumers to engage in physical activity was highlighted by CMO leaders as an important support strategy.


*Our support workers actually will attend gyms and classes with people, because we realise that it’s not only about that, sometimes people lack confidence, especially if they’re overweight, or they don’t have great self-esteem. So, people will accompany people to gyms and to classes.*
CMO leader 1


*We support people to engage to the best of their ability in regular exercise just throughout their routine.*
CMO leader 5

Whilst to a lesser extent, staff encouragement was also discussed in relation to smoking, nutrition, alcohol, and sleep by one or more CMO leaders.


*More recently… every staff member is expected to be interested in and trying to encourage a change in smoking behaviour.*
CMO leader 3


*If the consumer says [in the course of] the work with a staff member, I’m not sleeping, it’s a problem, the staff member will work with…and encourage the person to see their GP, do some basic environmental changes that are fairly widely accepted as common knowledge like coffee, don’t watch television in bed, darkened room, go to bed at the same time and all that sort of stuff.*
CMO leader 3

Peer support was discussed in relation to most health behaviours and overall support more generally. CMO leaders placed a high level of importance on the impact peer support had on consumers.


*We had a peer health coaching program [in relation to smoking]...where we would have consumers providing health coaching over the telephone to other consumers.*
CMO leader 3


*That relatability and a member of [name of organisation] works really hard on accessibility and being relatable for young people... I think that the peer workforce really takes that to a whole other level of hey, we’re doing this together. I understand where you’ve been and I’m here to do this with you.*
CMO leader 6

### 3.2. What Allows CMOs to Address SNAPS Behaviours for Consumers with a Mental Health Condition (Barriers and Facilitators)

This theme was often related to how CMOs are structured and the type of care they provide to consumers. For CMOs providing live-in facility/support, for instance, staff might be more aware of the sleep patterns of consumers and supports available to address these behaviours, compared to a CMO that provides a 2 h drop-in service in the middle of the day.

Some CMO leaders who worked with consumers that utilised drop-in services or short stay services commented on the additional complexities in providing SNAPS supports to consumers during contacts of shorter length; compared to opportunities afforded by longer term contact. CMO leaders of short stay services felt the reduced length of engagements with consumers impeded their ability to support long term health behaviour change.


*But we’ve got other drop-in support people who, outside of the hours that we provide services to, well, they go out in the community, they go to the pub, they go to the bottle shop and buy beer and whatever they want, their choice of drink and they bring it back to [unclear] and we don’t have a lot of input into that.*
CMO leader 5


*Our mandate really is to provide the very best experience we can in those five days for people… So, I think as an organisation, we provide the education and the one-on-one time with the counsellor but then it’s the follow up of course which is so important with smoking and drinking cessation that we don’t do.*
CMO leader 7

#### 3.2.1. Lack of Funding

A particularly salient barrier identified by CMO leaders was limited and inconsistent funding. For many CMOs, the capacity to provide SNAPS supports was dependent on the individual CMO’s funding model. For example, some CMO leaders discussed a combination of government funding, donations, research grants, funding specifically allocated to their consumer(s) (e.g., NDIS funds), and various opportunities to fund programs; while, other CMO leaders spoke only of government funding. Some CMOs only addressed SNAPS if these risk behaviours were acknowledged (and consequently funded) as requiring support in a consumer’s personalised NDIS care management plan. Other CMOs indicated they had additional funding to provide these SNAPS supports to consumers, regardless of whether or not this was consistent with an individual’s tailored health behaviour plan from the NDIS.


*I’m always going to say funding because funding should never stop you doing these incredibly important things, but it is sometimes the requirement in order to have a really [sustained] approach... I think I’d like to see funders, perhaps in terms of the funding we do get, be a little bit more open to us employing different types of people... It’d be great if I could turn say three or four of those positions into…personal trainer or nutritionist…but the problem is the way they fund this, we have to report a certain amount of activity and only certain types of activity is allowed to be counted.*
CMO leader 3

CMO leaders expressed their concern about the limited funding for SNAPS supports, particularly in regional areas, and the way funding was required to be spent.


*But the biggest challenge for me is when you’ve got someone that needs help, and particularly when you’ve got someone that’s willing to engage in help. That’s a really critical window, and what I’ve noticed is if you’ve got a very vulnerable person at a stage where they’re able to accept help, and you can’t get them that help, that’s a massive failing. But it’s a reality of the system that we can’t [all live in], particularly from a regional lens, who get less resources, less funding….So there’s all these extra, additional barriers, reductions in funding—not enough funding to begin with period, uncertain funding….*
CMO leader 8

#### 3.2.2. Perceived Limited Ability to Influence Behaviours

Some health behaviours were seen as lower priorities to address: for example, some leaders discussed sleep as appearing to have less importance than the other health behaviours and needing more direct attention broadly within society such as through education and awareness campaigns and health promotion.


*I think it’s pretty underrepresented in the health promotion ads. Like you regularly see stuff about smoking and alcohol, drink driving, and all those sorts of things, but the promotion of healthy sleep is probably pretty a bland topic in health promotion and probably wouldn’t get a lot of funding.*
CMO leader 11

Sometimes a particular health behaviour, and smoking in particular, was seen as beyond the capacity of CMO staff to modify.


*These guys are very stuck in their ways with smoking. It is super challenging.*
CMO leader 4


*So it doesn’t matter how much you put a smoking plan in or even with a restrictive practice, if the consumer has enough independence where they’re allowed in the community without the need for any additional one-on-one support, then all they need to do is withdraw money, sit on a bench and smoke a couple of cigarettes and that’s it.*
CMO leader 11

#### 3.2.3. Workplace Policies and Culture 

A main facilitator discussed by CMO leaders was workplace policies and culture that modelled positive health behaviours and encouraged staff to adopt such behaviours themselves. CMO leaders noted that upper management level staff promoted SNAPS behaviours through workplace initiatives, policies, and culture. Staff who directly engaged with consumers were encouraged to participate in and promote these behaviours, which had a direct and positive impact on consumers and provision of care for SNAPS.


*My experience is when you get staff to be part of that, take ownership of that, be accountable for that and make positive changes themselves, the cascading effect is better and greater when you’re dealing with people in the community that you’re serving.*
CMO leader 8


*We have a smoking policy for staff and we also have a smoking policy for the people that we support. That also covers illicit substances and alcohol as well. So that pretty much provides the main guide for staff as to what to do and what’s permitted, what’s not permitted, what the scope is.*
CMO leader 10

#### 3.2.4. Collaboration with Available Supports

CMO leaders discussed the importance of collaboration with other local organisations and professionals as a facilitator to providing support for SNAPS. The collaboration between CMOs and existing services provides the opportunity for shared knowledge, resources and to build a network of experienced SNAPS professionals.


*Regularly inviting people to come from local health districts and other providers in the community, to share information, education, and health promotions. We do things like the health promotions talk about things like—particularly around Christmas and holiday periods, a lot of the services focus on safe drinking and limiting drinking.*
CMO leader 2


*We might get the [psychiatric medication] nurse out, and that way we can make sure that new staff members get that relevant information, but also, it never hurts to refresh—for anyone to have a refresh on those types of issues. I think that’s then that little bit of collaboration with the local health district as well, where they don’t—they know that we work quite closely with them, so they’re happy to share that information with us.*
CMO leader 4

#### 3.2.5. Staff Training and Education

CMO leaders discussed the importance of educating and training all CMO staff about SNAPS as another facilitator to providing care for SNAPS to consumers. In particular, targeted SNAPS training enabled CMO staff to provide tailored support to their consumers’ needs.


*To be able to deliver really targeted support, train the staff, development of resources that will be useful to people accessing the service, and relevant to them and their needs.*
CMO leader 2


*So by training our staff and getting our staff to go through it, we’re sort of unlocking the lived experience that’s already there around some of these issues. When we’re able to do that, which is not everywhere, does enhance what we deliver because people are then able to say well, I did this.*
CMO leader 3


*We do a lot of internal training…any time any local training comes up that we can put staff through.*
CMO leader 9

## 4. Discussion

This study is the first to comprehensively explore via qualitative methodology: (1) the type of support provided to address five key health risk behaviours (smoking, poor nutrition, alcohol consumption, inadequate physical activity, and poor sleep) for CMO consumers with a mental health condition; and (2) organisational and staff level barriers and facilitators to providing this support. The study found that CMOs are providing consumer support across all SNAPS health risk behaviours in diverse ways. Three clear sub-themes were identified for how this support is provided: through practical education and life skills, referral pathways and local connections, and staff encouragement and peer support. The care provided for health risk behaviours was perceived to be impacted by differences in organisational structure; however, five clear sub-themes highlighted common barriers and facilitators among CMOs providing this support. These were: lack of funding, perceived limited ability to influence behaviours, workplace culture, collaboration with available supports, and staff training and education. Provision of support is variable and influenced by the type of service (e.g., drop-in vs. live-in services) and therefore length of opportunities for support (i.e., short or long term), whether requests for support are consumer-led, and availability of NDIS funding for support related to health risk behaviours.

The findings from this study that practical education, referral pathways, and peer-delivered support are strategies used by CMOs to address SNAPS among consumers with mental health conditions are similar to those reported in previous studies [8,15,36,37]. The knowledge that CMOs are already utilising these strategies to some degree is encouraging as they correspond to findings from a recent review of international evidence sources [8] regarding the type of strategies that might be effective in improving outcomes related to consumer physical health in mental health CMO settings. The review identified: education and advice (e.g., personalised fitness and nutrition plans), free lifestyle aids (e.g., Nicotine Replacement Therapy (NRT) and gym memberships), counselling and coaching (e.g., Quitline counselling), and assistance, practical support and demonstrations (e.g., hands-on nutrition activities including assisted grocery trips) to have positive impacts on consumer physical health outcomes. The review also identified that mental health peer-workers (i.e., support workers who use their own lived experience of a mental health condition purposively when working with consumers) may be important providers of SNAPS supports. In line with recommendations from the review, additional high-quality research studies co-developed with CMOs are needed to evaluate the optimal combination of type of provider and strategies to improve consumer SNAPS behaviours; and subsequently reduce the burden of chronic disease for this population.

Our findings identified that workplace culture and collaboration with available supports facilitated the provision of preventive care for health risk behaviours in CMOs. These findings align with those of Hanusaik et al. (2015) which investigated organisational factors that facilitated chronic disease preventive care among Canadian organisations (including both government-run and community managed organisations) mandated to provide primary prevention of chronic disease. Hanusaik et al. proposed that determinants of capacity to provide preventive care at an organisational level include identification of chronic disease prevention as a government and public priority, as well as access to external resources and supports, and that internal policies and workplace supports were also influential [38]. Likewise, in research conducted in comparable mental health community settings in the US, the provision of care to broadly address the physical health of people with mental health conditions has been noted to align with internal organisational policies (‘strong organisational fit’), although this could be hindered by factors such as inadequacies of electronic patient record systems and difficulties coordinating with external providers [24]. These findings suggest that including the delivery of preventive care to mental health consumers in organisational guidelines and policies, and having access to both external and internal resources, may be important for supporting CMOs to address health risk behaviours among people with mental health conditions.

Further, findings from our current study, as well as past evidence [7,16,17], support the importance of factors such as staff training and education and positive workplace culture in facilitating the provision of preventive care for health risk behaviours. However, in practice, it is unclear whether policies to promote workplace culture or staff education and training are routinely and systematically embedded in CMOs. Our previous research among CMO leaders in NSW found that only 12% of CMOs had written policies or guidelines for all SNAPS behaviours, and only 40% had staff that had received training to address all SNAPS behaviours [7]. Future research could explore whether increasing routine and systematic provision of capacity-building opportunities in CMOs, such as regular training specifically aimed at providing preventive care for all SNAPS behaviours, facilitates an increase in care to address health risk behaviours among CMO consumers.

The findings that a lack of funding and a perceived limited ability to influence health risk behaviours were found as barriers to the provision of preventive care were perhaps not unexpected given the funding and administrative context within which CMOs operate. The diversity of funding CMOs receive, which can be temporary (e.g., grants), program-specific (e.g., HASI), or consumer-directed (e.g., NDIS), was described by participants as directly impacting whether care for health risk behaviours was provided. For example, for CMO consumers who have a NDIS plan [30,32], the receipt of support to address health risk behaviours may depend on whether it is included in their approved care management plan. Further, inflexible funding arrangements, with constraints regarding the type of activity and staff roles able to be funded, were noted as impeding CMOs ability to employ physical health professionals (e.g., nutritionists). The financial position of CMOs may also impact the diversity of supports provided, as well as the variable prioritisation of care for different health risk behaviours. The inclusion of preventive care to address SNAPS behaviours as part of CMO funding agreements or individual consumer care management plans may be one potential strategy to increase the ability of CMOs to support consumers to address their health risk behaviours. Beyond funding, various factors may impact CMOs perceived limited ability to influence health risk behaviours, including the length of time spent with consumers and perceptions of consumer disinterest or inability to make changes to behaviours, particularly smoking. Additional strategies such as peer champions (employed in stand-alone roles such as a ‘peer health coach’) [36], consumer narratives [39], and information provided to CMO staff regarding consumer interest in, and efficacy of behaviour change supports (e.g., NRT) [15] may be needed to improve CMOs’ perceived ability to influence health behaviours among consumers.

The findings of this study should be considered in light of some of the strengths and limitations of its design and methodology. A strength of the research was the rigour of the inductive approach to analysis, for example: coders kept a reflexive audit of any pre-existing assumptions and biases throughout the coding process, noting these at the end of all coding sessions [40]; and coding was led by one senior coder and two secondary coders, and all interviews were double coded [41]. In contrast, response rate and ‘level of informant’ should be noted as potential limitations of the study. Although data saturation was reached with 12 informants, it cannot be assumed that findings are generalisable to all CMOs operating in NSW. Further, the interviews were conducted with staff in senior management and leadership roles in participating CMOs. Although some staff in these positions had dual roles of management and on the ground provision of support, it is possible that not all barriers that frontline staff experience in delivering SNAPS supports or aids were identified by all informants. Further interviews could be conducted with staff who only deliver frontline support, to investigate whether any additional barriers or facilitators are identified. Relatedly, an important direction for future research will be to incorporate the views of consumers and/or consumer representatives. Finally, this qualitative study was conducted during the COVID-19 pandemic, and its findings need to be considered within this context. During interviews, informants were asked to only reflect on usual service provision rather than adaptations made to care in order to adhere to COVID-19 rules and regulations regarding social distancing (e.g., transitioning from face-to-face to digital supports). Some CMO leaders did discuss the impact of COVID-19; however, such content was not coded nor analysed as it was not an aim of the study to capture COVID-19 impact.

## 5. Conclusions

This study provides an in-depth exploration of the type of support provided by CMOs to address the health risk behaviours of consumers living with mental health conditions, and organisational and staff level barriers and facilitators to providing such support. Many CMOs have organisational cultures that are highly supportive of health behaviour support provision; however, doing so is variable across SNAPS behaviours and currently hindered by factors including funding and perceived limited ability to influence behaviours. The study identified a number of practical implications regarding areas of priority for increasing the delivery of SNAPS support for consumers, including: funding at an organisational level to ensure support is provided to all consumers for all health risk behaviours; staff education and training in relation to SNAPS; and embedding preventive care for health risk behaviours both in internal and external guidelines, policies, and funding in order to better align internal and external factors, and see sustained improvements in care provision in this setting. Future research exploring consumer experiences of such care provision and studies utilising rigorous study designs (i.e., randomised controlled trials) to evaluate the effectiveness of such care in achieving consumer outcomes is needed to fully realise the potential of CMOs in improving the health risk behaviours of people living with mental health conditions.

## Figures and Tables

**Figure 1 ijerph-19-05533-f001:**
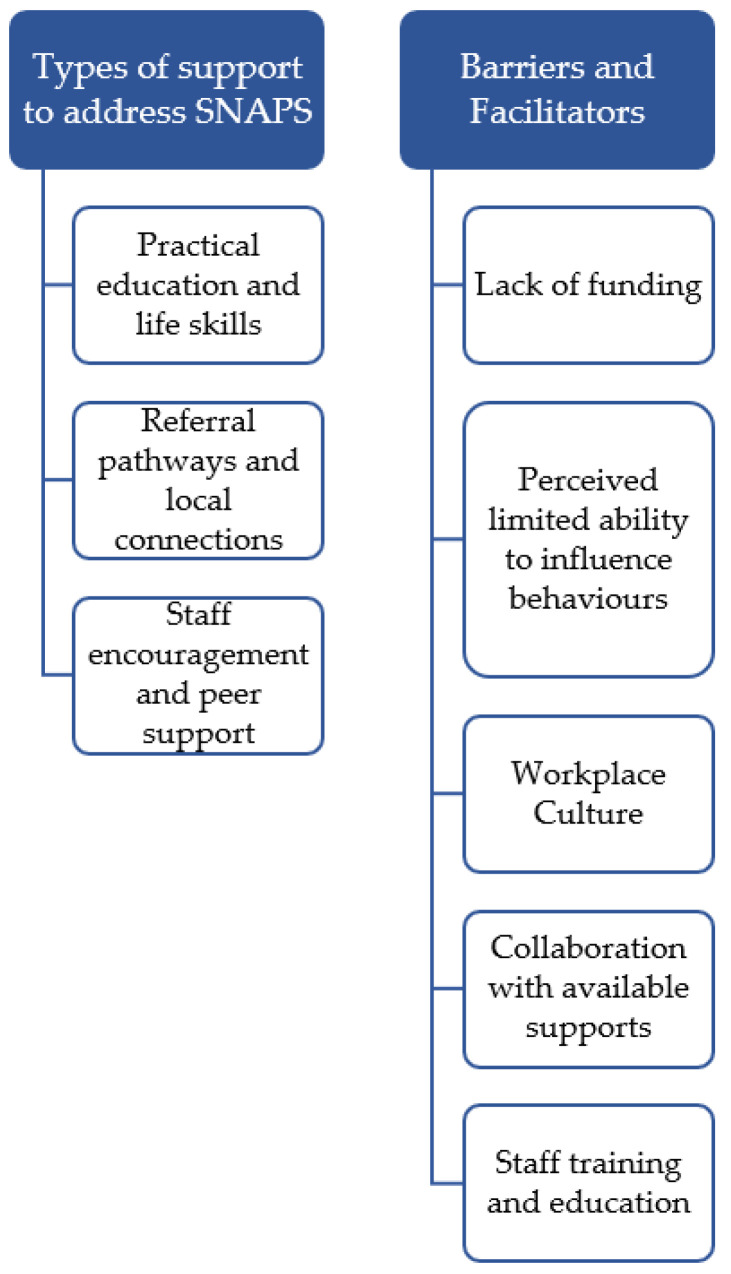
Theme map.

## Data Availability

Restrictions apply to the availability of these data. The data are not publicly available due to protecting the confidentiality of study participants where data contains easily identifiable information.

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
