# Peer review of "Exploring Support Provided by Community Managed Organisations to Address Health Risk Behaviours Associated with Chronic Disease among People with Mental Health Conditions: A Qualitative Study with Organisational Leaders"

_ijerph, 2022, doi:10.3390/ijerph19095533_

Round 1

Reviewer 1 Report

Mental diseases are a prevalent pathology of high prevalence that involve expensive consequences in quality of life to those who suffer from being associated with risk behaviors and chronic diseases. This research explores types of organizational organizations and strategies to follow to help affected people.

Thus, work addresses a topic of considerable interest. In addition, it is well justified and well written.

From my opinion, its weakness, not avoidable, lies in its exploratory nature. Future investigations with controlled random designs that study the effects of such assistance benefits are needed.

In my opinion, the paper evaluated can be accepted for publication.

Author Response

Response to Reviewer 1 Comments

Point 1: Mental diseases are a prevalent pathology of high prevalence that involve expensive consequences in quality of life to those who suffer from being associated with risk behaviors and chronic diseases. This research explores types of organizational organizations and strategies to follow to help affected people.

Thus, work addresses a topic of considerable interest. In addition, it is well justified and well written.

From my opinion, its weakness, not avoidable, lies in its exploratory nature. Future investigations with controlled random designs that study the effects of such assistance benefits are needed.

Response 1: Thank you for your suggestion. We agree this is an important direction of future work, the identification of which has been retained in the following sentence of the discussion section of the paper:

“Future research exploring consumer experiences of such care provision and studies utilising rigorous study designs (i.e. randomised controlled trials) to evaluate the effectiveness of such care in achieving consumer outcomes is needed to fully realise the potential of CMOs in improving the health risk behaviours of people living with mental health conditions.” (See page 16)

Point 2: In my opinion, the paper evaluated can be accepted for publication.

Response 2: Thank you for reviewing and evaluating our publication.

Reviewer 2 Report

In this article the authors address an extremely interesting topic through interviews. The introduction is well written and provides insight into the topic. The methodology is correct and well thorough. The arguments are exhaustive. In my opinion, the authors in the final part of the article shouldbetter describe the practical implications of the study and limitations. In conclusion, the article is interesting but needs a revision to be published in the Journal.

in my opinion the authors should better describe the practical perspectives of the study. What can this article improve in clinical practice? What are the practical aims? They should also describe the limitations. Furthermore, the Authors must modify the style of the references as indicated by the Journal.

Author Response

Response to Reviewer 2 Comments

Point 1: In this article the authors address an extremely interesting topic through interviews. The introduction is well written and provides insight into the topic. The methodology is correct and well thorough. 

The arguments are exhaustive. In my opinion, the authors in the final part of the article should better describe the practical implications of the study and limitations. In conclusion, the article is interesting but needs a revision to be published in the Journal. in my opinion the authors should better describe the practical perspectives of the study. What can this article improve in clinical practice? What are the practical aims? 

Response 1: Thank you for reviewing the manuscript and providing suggestions for improvements. As per your recommendation, the ‘conclusion’ section of the paper has been updated to better describe the practical implications of the study (page 16):

“The study identified a number of practical implications regarding areas of priority for increasing the delivery of SNAPS support for consumers, including: funding at an organisational level to ensure support is provided to all consumers for all health risk behaviours; staff education and training in relation to SNAPS; and embedding preventive care for health risk behaviours both in internal and external guidelines, policies and funding in order to better align internal and external factors, and see sustained improvements in care provision in this setting.”

Point 2: They should also describe the limitations.

Response 2: The limitations of the study have been outlined in the final paragraph (page 15) of the discussion:

“The findings of this study should be considered in light of some of the strengths and limitations of it’s design and methodology. A strength of the research was the rigour of the inductive approach to analysis, for example: coders kept a reflexive audit of any pre-existing assumptions and biases throughout the coding process, noting these at the end of all coding sessions [40]; and coding was led by one senior coder and two secondary coders, and all interviews were double coded [41]. In contrast, response rate and ‘level of informant’ should be noted as potential limitations of the study. Although data saturation was reached with 12 informants, it cannot be assumed that findings are generalisable to all CMOs operating in NSW. Further, the interviews were conducted with staff in senior management and leadership roles in participating CMOs. Although some staff in these positions had dual roles of management and on the ground provision of support, it is possible that not all barriers that frontline staff experience in delivering SNAPS supports or aids were identified by all informants. Further interviews could be conducted with staff who only deliver frontline support, to investigate whether any additional barriers or facilitators are identified. Relatedly, an important direction for future research will be to incorporate the views of consumers and/or consumer representatives. Finally, this qualitative study was conducted during the COVID-19 pandemic, and it’s findings need to be considered within this context. During interviews, informants were asked to only reflect on usual service provision rather than adaptations made to care in order to adhere to COVID-19 rules and regulations regarding social distancing (e.g., transitioning from face to face to digital supports). Some CMO leaders did discuss the impact of COVID-19, however such content was not coded nor analysed as it was not an aim of the study to capture COVID impact.”

Point 3: Furthermore, the Authors must modify the style of the references as indicated by the Journal.

Response 3: Finally, we have aligned our style of referencing with that outlined in the IJERPH template for authors.

Reviewer 3 Report

Dear Authors,
Thank you for submitting the paper to the IJERPH journal. Your research topic is compatible with the aims and scopes of the journal and special issue. All the elements which should be indicated in a research paper, were addressed. The structure is clear, and the main aspects of the research framework were mentioned. The procedure of collecting the qualitative data was mentioned.

The paper is well prepared, although I would like to advise you on some way to improve it. In my opinion, a significant improvement would be to include the table with the main characteristics of interviewed CMO leaders, like age, gender, and experience at work in CMO generally and as a leader (even if not in detail but at some intervals), maybe main tasks or functions. 

I have no additional comments on the remaining text of the manuscript. I think it is publishable.

Technical issues:
Please check all the technical aspects of your manuscript, if they are compatible with the journal's requirements.

Author Response

Response to Reviewer 3 Comments

Point 1: Dear Authors, Thank you for submitting the paper to the IJERPH journal. Your research topic is compatible with the aims and scopes of the journal and special issue. All the elements which should be indicated in a research paper, were addressed. The structure is clear, and the main aspects of the research framework were mentioned. The procedure of collecting the qualitative data was mentioned.

The paper is well prepared, although I would like to advise you on some way to improve it. In my opinion, a significant improvement would be to include the table with the main characteristics of interviewed CMO leaders, like age, gender, and experience at work in CMO generally and as a leader (even if not in detail but at some intervals), maybe main tasks or functions. 

Response 1: Thank you for your feedback. Unfortunately, the demographic characteristics of the participating CMO leaders were not obtained during the study. However, the following information regarding the position of the leader within the organisation has been added to the results section of the manuscript (Page 5). Please note each organisation had specific names for the positions or roles leaders were employed within and therefore to maintain the confidentiality of participants the positions have been categorised broadly.

“Two participants were employed as Chief Executive Officers and the remaining ten participants were employed in other senior management roles (e.g., ‘NSW State Manager’ and ‘Executive Manager, Operations’).”

Point 2: I have no additional comments on the remaining text of the manuscript. I think it is publishable.

Response 2: Thank you for reviewing the manuscript and support for publication.

Point 3: Technical issues:
Please check all the technical aspects of your manuscript, if they are compatible with the journal's requirements.

Response 3: Thank you, we have checked the technical aspects of the manuscript against the journal’s requirements and adjusted where needed.

Reviewer 4 Report

I have carefully read the article which I consider very interesting for an international audience. The topic is certainly important and I consider it worthy of publication.
However, I think some points are needed to improve:
- In the introduction one could also briefly insert the concept that community adoption for this type of patient also helps in keeping risks under control, those publication could be cited:

doi: 10.1186 / s12913-018-3846-7.

doi: 10.1080 / 15563650.2017.1334913.

doi: 10.3390 / ijerph18073425.
- Study limits and possibilities for future studies should be included
- The bibliography should be corrected according to editorial standards and possibly extended to other international contributions relevant to the topic

- The authors could then clearly indicate the time period analyzed and the origin of the data.

Author Response

Response to Reviewer 4 Comments

Point 1: I have carefully read the article which I consider very interesting for an international audience. The topic is certainly important and I consider it worthy of publication.

However, I think some points are needed to improve:
- In the introduction one could also briefly insert the concept that community adoption for this type of patient also helps in keeping risks under control, those publication could be cited:

doi: 10.1186 / s12913-018-3846-7.

doi: 10.1080 / 15563650.2017.1334913.

doi: 10.3390 / ijerph18073425.

Response 1: Thank you, we read these with interest. Unfortunately, the references refer to risk management, hazard exposure and care in medical and hospital settings which are settings that do not align with the setting of the current manuscript (i.e. community managed organisations). For this reason we do not believe they are appropriate for incorporation into the introduction of the manuscript.

Point 2:- Study limits and possibilities for future studies should be included

Response 2: The limitations of the study have been outlined in the final paragraph (page 15) of the discussion:

“The findings of this study should be considered in light of some of the strengths and limitations of it’s design and methodology. A strength of the research was the rigour of the inductive approach to analysis, for example: coders kept a reflexive audit of any pre-existing assumptions and biases throughout the coding process, noting these at the end of all coding sessions [40]; and coding was led by one senior coder and two secondary coders, and all interviews were double coded [41]. In contrast, response rate and ‘level of informant’ should be noted as potential limitations of the study. Although data saturation was reached with 12 informants, it cannot be assumed that findings are generalisable to all CMOs operating in NSW. Further, the interviews were conducted with staff in senior management and leadership roles in participating CMOs. Although some staff in these positions had dual roles of management and on the ground provision of support, it is possible that not all barriers that frontline staff experience in delivering SNAPS supports or aids were identified by all informants. Further interviews could be conducted with staff who only deliver frontline support, to investigate whether any additional barriers or facilitators are identified. Relatedly, an important direction for future research will be to incorporate the views of consumers and/or consumer representatives. Finally, this qualitative study was conducted during the COVID-19 pandemic, and it’s findings need to be considered within this context. During interviews, informants were asked to only reflect on usual service provision rather than adaptations made to care in order to adhere to COVID-19 rules and regulations regarding social distancing (e.g., transitioning from face to face to digital supports). Some CMO leaders did discuss the impact of COVID-19, however such content was not coded nor analysed as it was not an aim of the study to capture COVID impact.”

The possibilities for future studies have been outlined in the discussion and conclusion section of the manuscript:

“Future research could explore whether increasing routine and systematic provision of capacity-building opportunities in CMOs, such as regular training specifically aimed at providing preventive care for all SNAPS behaviours, facilitates an increase in care to address health risk behaviours among CMO consumers.” (Page 14).

“Future research exploring consumer experiences of such care provision and studies utilising rigorous study designs (i.e. randomised controlled trials) to evaluate the effectiveness of such care in achieving consumer outcomes is needed to fully realise the potential of CMOs in improving the health risk behaviours of people living with mental health conditions.” (Page 16).

Point 3:- The bibliography should be corrected according to editorial standards and possibly extended to other international contributions relevant to the topic

Response 3: The bibliography has been corrected according to the journal’s requirements.

Point 4:- The authors could then clearly indicate the time period analyzed and the origin of the data.

Response 4: Thank you for your comment. Information regarding the time-period of data collection for the study and how participants were recruited to the study is retained at the following locations within the manuscript:

Time period of data collection (Page 5): “From the 85 invited eligible CMOs, a total of twelve participants from twelve CMOs agreed to participate and completed interviews (14%) between March and July 2020.”

Origin of data (page 3 to 5): Information pertaining to the origin of the data has been outlined in the methods section of the manuscript, including how participants were identified and recruited to the study (see section ‘participants and recruitment’, page 4) and the process for collecting and transcribing data (see section ‘data collection’, page 5).

Round 2

Reviewer 4 Report

I have read the article carefully, I think it is of interest to an international audience and deserves to be published.
Small suggestions that the authors could insert:
- communities also include prisons, these are particular contexts that often also host patients with chronic diseases, it could be briefly mentioned and this work cited: doi: 10.3390 / healthcare9111511.
- the references that indicate a website should be reviewed according to the editorial indications of the journal
- in the contributions of the authors, only the initials should be placed and the e-mail addresses of each author should be inserted